# Taste quality and hunger interactions in a feeding sensorimotor circuit

Philip K Shiu[1†], Gabriella R Sterne[1,2*†], Stefanie Engert[1], Barry J Dickson[2,3], Kristin Scott[1*]

[1]University of California, Berkeley, Berkeley, United States; [2]Janelia Research Campus, Howard Hughes Medical Institute, Chevy Chase, United States; [3]Queensland Brain Institute, University of Queensland, Brisbane, Australia

**Summary** Taste detection and hunger state dynamically regulate the decision to initiate feeding. To study how context-appropriate feeding decisions are generated, we combined synaptic resolution circuit reconstruction with targeted genetic access to specific neurons to elucidate a gustatory sensorimotor circuit for feeding initiation in adult *Drosophila melanogaster*. This circuit connects gustatory sensory neurons to proboscis motor neurons through three intermediate layers. Most neurons in this pathway are necessary and sufficient for proboscis extension, a feeding initiation behavior, and respond selectively to sugar taste detection. Pathway activity is amplified by hunger signals that act at select second-order neurons to promote feeding initiation in food-deprived animals. In contrast, the feeding initiation circuit is inhibited by a bitter taste pathway that impinges on premotor neurons, illuminating a local motif that weighs sugar and bitter taste detection to adjust the behavioral outcomes. Together, these studies reveal central mechanisms for the integration of external taste detection and internal nutritive state to flexibly execute a critical feeding decision.

*For correspondence:
sternegr@berkeley.edu (GRS);
kscott@berkeley.edu (KS)

†These authors contributed equally to this work

Competing interest: The authors declare that no competing interests exist.

## Editor's evaluation

The findings presented in this article contribute to a circuit-based understanding of how sweet and bitter tastes are integrated with the hunger state to drive feeding initiation in *Drosophila*. Anatomical, behavioral, and neuronal activity data support a multi-step pathway from sensory input to motor output. This manuscript, thus, advances our understanding of how multiple sensory cues are integrated with an internal state to reach a behavioral decision.

## Introduction

The decision to initiate feeding depends both on the quality of available food and current nutrient needs. The gustatory system detects nutritious and noxious compounds in the environment and evaluates food quality. Food quality information is integrated with internal nutritive state to ensure that food intake matches energy demands. How do central neural circuits evaluate taste information in the context of internal nutritive state to make feeding decisions?

As feeding decisions are universal and essential for survival, animals as diverse as humans and *Drosophila* share similar strategies to detect taste compounds and assess nutrient needs. Peripheral taste detection in mammals and insects is mediated by sensory cells that detect specific taste modalities and elicit innate feeding behaviors. Both mammals and flies have sugar-, bitter-, water-, and salt-sensing gustatory cells (*Liman et al., 2014*; *Yarmolinsky et al., 2009*). Activation of sugar-sensing gustatory cells triggers feeding initiation, whereas activation of bitter-sensing cells inhibits feeding. Mammals and insects also evaluate internal nutrient needs with similar strategies (*Augustine et al.,*

*2018*; *Leopold and Perrimon, 2007*; *Nässel and Zandawala, 2019*; *Pool and Scott, 2014*; *Sareen et al., 2021*). Neuromodulators released from neurosecretory centers and the gut signal hunger or satiety to oppositely regulate feeding. Disruption of these hunger and satiety signals results in obesity and anorexia in mammals and insects.

Although gustatory sensory neurons have been shown to be modulated by hunger signals and conflicting taste information (*Chu et al., 2014*; *French et al., 2015*; *Inagaki et al., 2012*; *Inagaki et al., 2014*; *Jeong et al., 2013*; *Meunier et al., 2003*), central mechanisms that modulate feeding decisions are unclear because the identity, structure, and function of central feeding initiation circuits are unknown. Recent advances in brain-wide synaptic connectivity mapping (*Dorkenwald et al., 2022*; *Eckstein et al., 2020*; *Zheng et al., 2018*) and precise genetic access to single neurons (*Dionne et al., 2018*; *Luan et al., 2006*) make *Drosophila melanogaster* an ideal system to interrogate how central neural circuits compute feeding decisions. Taste detection in adult *Drosophila* begins with activation of gustatory receptor neurons (GRNs) found in sensory structures located on the body surface, including the external mouthparts, or proboscis labellum (*Dethier, 1976*; *Montell, 2021*; *Scott, 2018*; *Stocker, 1994*). The axons of proboscis GRNs project to the primary taste and premotor center of the insect brain, the subesophageal zone (SEZ) (*Kendroud et al., 2018*; *Miyazaki and Ito, 2010*; *Stocker and Schorderet, 1981*). The motor neurons that execute feeding have cell bodies and dendrites in the SEZ near GRN axons, suggesting a local feeding circuit (*Gordon and Scott, 2009*; *McKellar et al., 2020*; *Schwarz et al., 2017*). However, only a few isolated interneurons have been implicated in feeding initiation (*Flood et al., 2013*; *Kain and Dahanukar, 2015*).

To investigate how neural circuits transform taste detection into context-appropriate feeding decisions, we combined electron microscopy-based circuit reconstruction, genetic tools that provide access to single cell types, optogenetics, and imaging of taste responses in awake, behaving animals to uncover a circuit-level view of feeding initiation in *Drosophila melanogaster*. This work delineates the neural circuit that transforms taste detection into the motor actions of feeding initiation from sensory inputs to motor outputs, and reveals central mechanisms that integrate taste detection with internal physiological state to shape behavior.

## Results

### GRNs synapse onto multiple second-order neurons

To examine neural circuits for feeding initiation, we identified neurons directly postsynaptic to gustatory sensory axons in the central brain. We utilized the full adult fly brain (FAFB) electron microscopy (EM) volume (*Zheng et al., 2018*) to manually reconstruct neurons postsynaptic to 17 labellar GRN axons in the right hemisphere that likely correspond to sugar-sensing GRNs (*Engert et al., 2021*). Fifteen second-order taste neurons and their synapses were fully reconstructed (*Figure 1A–B* and *Figure 1—figure supplement 1*) in the CATMAID platform (*Saalfeld et al., 2009*). The second-order neurons do not receive inputs from all candidate sugar GRNs (*Figure 1—figure supplement 1B*); however, as sugar GRNs make extensive axonal-axonal connections (*Engert et al., 2021*), the impact of these differences is unclear.

To assess the completeness of our second-order collection, we compared these 15 second-order neurons with the recently released Flywire dataset, a dense, machine learning based reconstruction of FAFB neurons (*Dorkenwald et al., 2022*; *Eckstein et al., 2020*). This comparison revealed that we identified 14 of the 15 neurons with the most synapses from sugar GRNs. These second-order neurons represent 12 unique cell types (*Figure 1—figure supplement 1F*). The 15 second-order neurons we manually reconstructed receive 21% of sugar GRN synaptic outputs. We note that the distribution of second-order neurons has a very long tail, likely due in part to small neural fragments that are challenging to reconstruct. These second-order neurons have not previously been characterized, except for G2N-1, which was identified as a candidate second-order gustatory neuron based on anatomical proximity to sugar-sensing GRNs (*Miyazaki et al., 2015*). Each of the second-order neurons is a local SEZ interneuron with arbors that overlap extensively with sugar GRN termini.

### Multiple second-order taste neurons influence proboscis extension

To test whether second-order gustatory neurons participate in feeding behaviors, we identified split-Gal4 lines that provide specific genetic access to individual second-order cell types, using

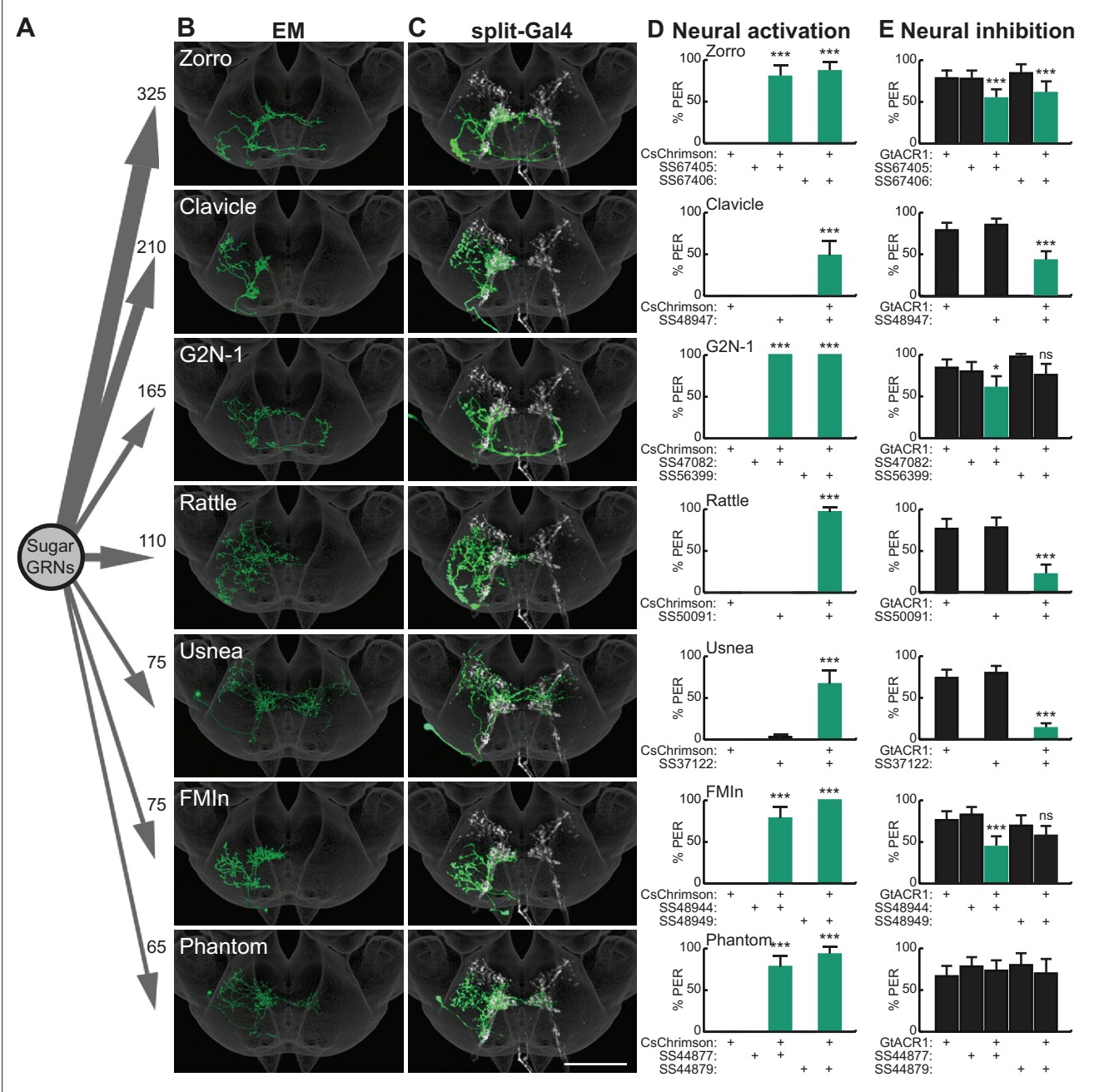

**Figure 1.** Sugar-sensing gustatory receptor neurons (GRNs) synapse onto multiple second-order neurons that influence proboscis extension. (**A**) Aggregate synaptic connectivity from sugar GRNs onto second-order sugar neurons. Numbers indicate the total number of synapses that the 17 candidate sugar GRNs make onto each second-order neuron. (**B–C**) Manually reconstructed electron microscopy (EM) skeletons (**B**) and registered neural images in split-Gal4 lines (**C**) for each second-order neuron in the subesophageal zone (SEZ) of the *Drosophila* brain. Sugar GRNs are depicted in white, JRC 2018 unisex coordinate space is shown in gray (**C**). Scale bar is 50 µm. (**D**) CsChrimson-mediated activation of seven second-order neurons elicits proboscis extension, n=30 flies per genotype. (**E**) GtACR1-mediated inhibition of second-order neurons reduces proboscis extension to 50 mM sucrose, n=46–83 flies per genotype. (**D–E**) The fraction of flies exhibiting proboscis extension response (PER) upon optogenetic or 50 mM sucrose stimulation. Mean ± 95% confidence interval (CI), Fisher's Exact Tests, *p<0.05, ***p<0.001. See **Figure 1—figure supplement 1** for EM reconstructions of additional second-order neurons and synaptic connectivity counts. See **Figure 1—figure supplement 2** for additional PER phenotypes of second-order sugar neurons.

The online version of this article includes the following source data and figure supplement(s) for figure 1:

**Source data 1.** Source data for behavioral experiments in **Figure 1**.

**Figure supplement 1.** Anatomy of reconstructed second-order neurons and their connectivity, related to **Figure 1**.

*Figure 1 continued on next page*

eLife Research article

*Figure 1 continued*

**Figure supplement 2.** Additional proboscis extension phenotypes of second-order neurons, related to *Figure 1*.

**Figure supplement 2—source data 1.** Source data for behavioral experiments in *Figure 1—figure supplement 2*.

NBLAST comparisons (*Costa et al., 2016*) to a library of SEZ split-Gal4 lines (*Sterne et al., 2021*). This provided split-Gal4 matches for seven second-order neurons (*Figure 1C* and *Figure 1—figure supplement 2A*). Additionally, we used intersectional approaches to gain genetic access to two additional second-order neurons, Cleaver (*Figure 1—figure supplement 2A*) and Zorro (*Figure 1C*). These genetic reagents are exquisitely specific for each of the nine second-order gustatory neurons, providing the opportunity to evaluate their function.

As activation of sugar-sensing GRNs on the proboscis labellum causes the fly to extend its proboscis to initiate feeding (*Dethier, 1976*), we tested whether activation or inhibition of second-order taste neurons influences this behavior. We expressed the red-shifted channelrhodopsin CsChrimson (*Klapoetke et al., 2014*) selectively in each second-order taste neuron, activated each with 635 nm light, and examined the proboscis extension response (PER). Remarkably, optogenetic activation of seven of the nine second-order taste neurons elicited proboscis extension (*Figure 1D* and *Figure 1—figure supplement 2B*). Moreover, inhibiting the activity of each second-order neuron individually, by optogenetic activation of the anion channelrhodopsin GtACR1 (*Mohammad et al., 2017*), reduced proboscis extension to 50 mM sucrose in food-deprived flies, for six of the seven second-order neurons that elicited PER upon activation (*Figure 1E*). At a higher sucrose concentration (100 mM), neural inhibition of only two of the second-order neuron classes decreased proboscis extension (*Figure 1—figure supplement 2C*). These studies argue that multiple second-order neurons contribute to normal feeding initiation behavior and suggest that the partial redundancy of these second-order neurons ensures robust feeding.

## Second-order taste neurons activate a local SEZ circuit for feeding initiation

How does activation of a diverse set of second-order neurons drive proboscis extension? Proboscis motor neurons 4, 6, 7, and 9 are involved in extending different segments of the proboscis for feeding initiation (*McKellar et al., 2020*). We focused on the well-studied motor neuron 9 (MN9), which is necessary and sufficient for extension of the rostrum, the largest portion of the proboscis (*Gordon and Scott, 2009*; *McKellar et al., 2020*). We located MN9 in the FAFB EM volume by examining large SEZ neurons that lack synaptic output. To identify a pathway from taste detection to proboscis extension, we reconstructed presynaptic partners of MN9 and postsynaptic partners of second-order taste neurons.

This strategy identified a minimal pathway from taste detection to proboscis extension, composed of interconnected second-order neurons and third-order neurons each receiving inputs from a subset of second-order neurons, and feedforward premotor neurons (*Figure 2A* and *Figure 2—figure supplement 1A*). The third-order neurons represent a small subset based on comparisons to Flywire automated reconstructions (*Figure 2—figure supplement 1B*). They include one previously characterized neuron, the putative feeding command neuron, Fdg (*Flood et al., 2013*; *Figure 3B*), and a set of descending neurons, Bract, that project to the ventral nerve cord (*Sterne et al., 2021*). The premotor neurons are strongly connected to MN9, representing approximately 13% of the synaptic input onto MN9 (*Figure 2—figure supplement 1C*). There are direct connections between three second-order neurons (G2N-1, Zorro, and FMIn) and premotor neurons, and additional paths via third-order neurons to premotor neurons. The neurons in the feeding initiation circuit are predicted to be cholinergic, except for Usnea and Phantom which are predicted to be GABAergic (*Figure 2—figure supplement 1D*), based on machine learning classifications (*Eckstein et al., 2020*).

To investigate the function of deeper layers of this circuit, we identified split-Gal4 lines that selectively label two third-order neurons and one premotor neuron (*Figure 2C*) using NBLAST comparisons with SEZ split-Gal4 lines (*Sterne et al., 2021*). Optogenetic activation of third-order or premotor neurons with CsChrimson revealed that each cell type elicits robust proboscis extension (*Figure 2D*). However, acute inhibition of the third-order or premotor neurons with GtACR1 did not influence PER to 50 mM sucrose (*Figure 2E*), consistent with multiple pathways to proboscis motor neurons. Thus,

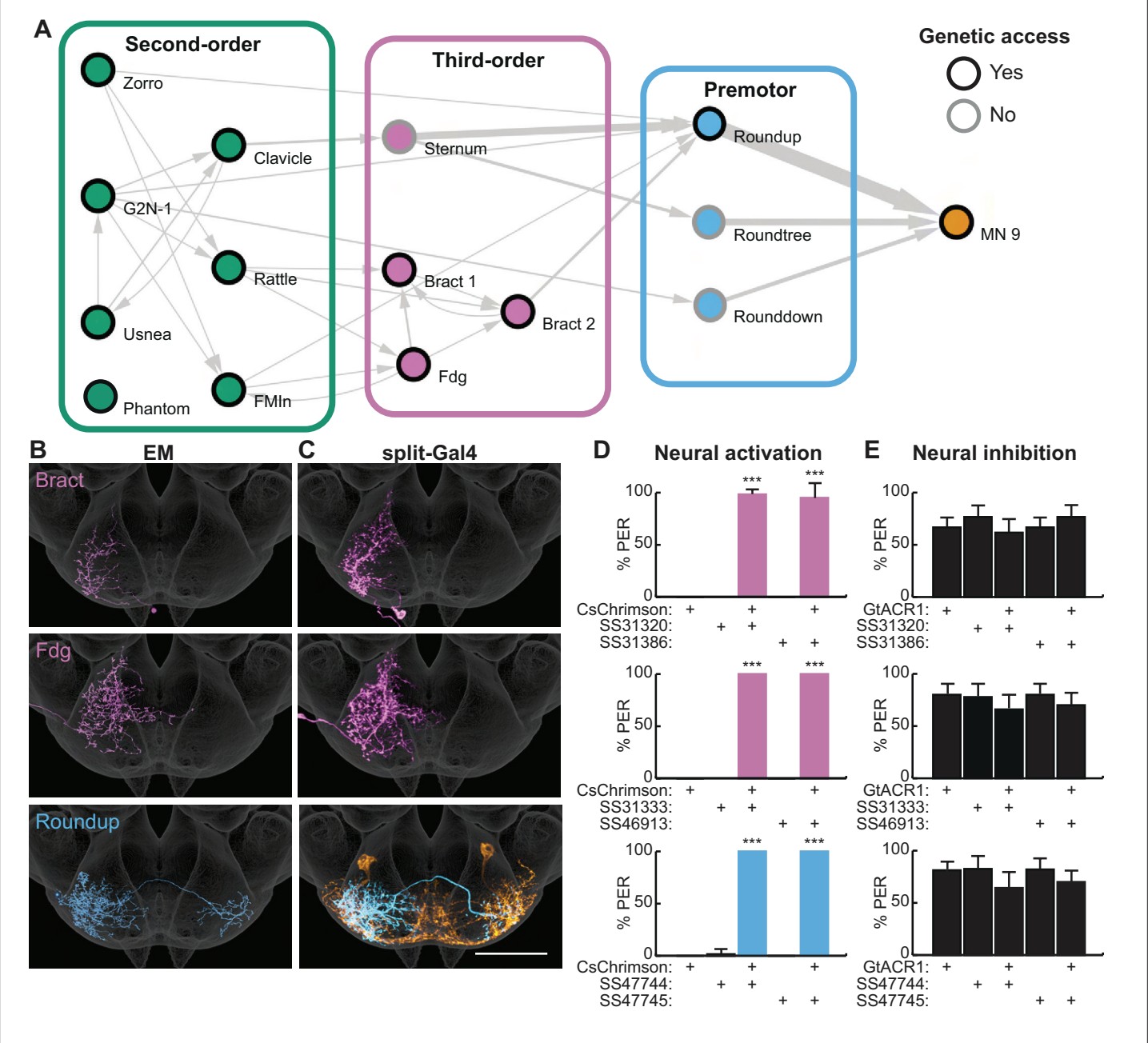

**Figure 2.** Second-order neurons synapse onto a local sensorimotor circuit for feeding initiation. (**A**) Schematic of the feeding initiation circuit. Circles outlined in black denote neurons with split-Gal4 genetic access, circles with gray outlines denote neurons without split-Gal4 genetic access. Line thickness represents synaptic connectivity of more than five synapses. (**B–C**) Electron microscopy (EM) neural reconstructions (**B**) and registered neural images in split-Gal4 lines (**C**) of third-order or premotor neurons in the subesophageal zone (SEZ). Scale bar is 50 μm. JRC 2018 unisex coordinate space is shown in gray, MN9 morphology is shown in orange. (**D**) CsChrimson-mediated activation of third-order or premotor neurons elicits proboscis extension response (PER), n=30 flies per genotype. (**E**) GtACR1-mediated inhibition of third-order or premotor neurons does not influence PER to 50 mM sucrose, n=40–70 flies per genotype. (**D–E**) Mean ± 95% CI, Fisher's Exact Tests, ***p<0.001. See *Figure 2—figure supplement 1* for synaptic counts.

The online version of this article includes the following source data and figure supplement(s) for figure 2:

**Source data 1.** Source data for behavioral experiments in *Figure 2*.

**Figure supplement 1.** Synaptic connectivity in the feeding initiation circuit, related to *Figure 2*.

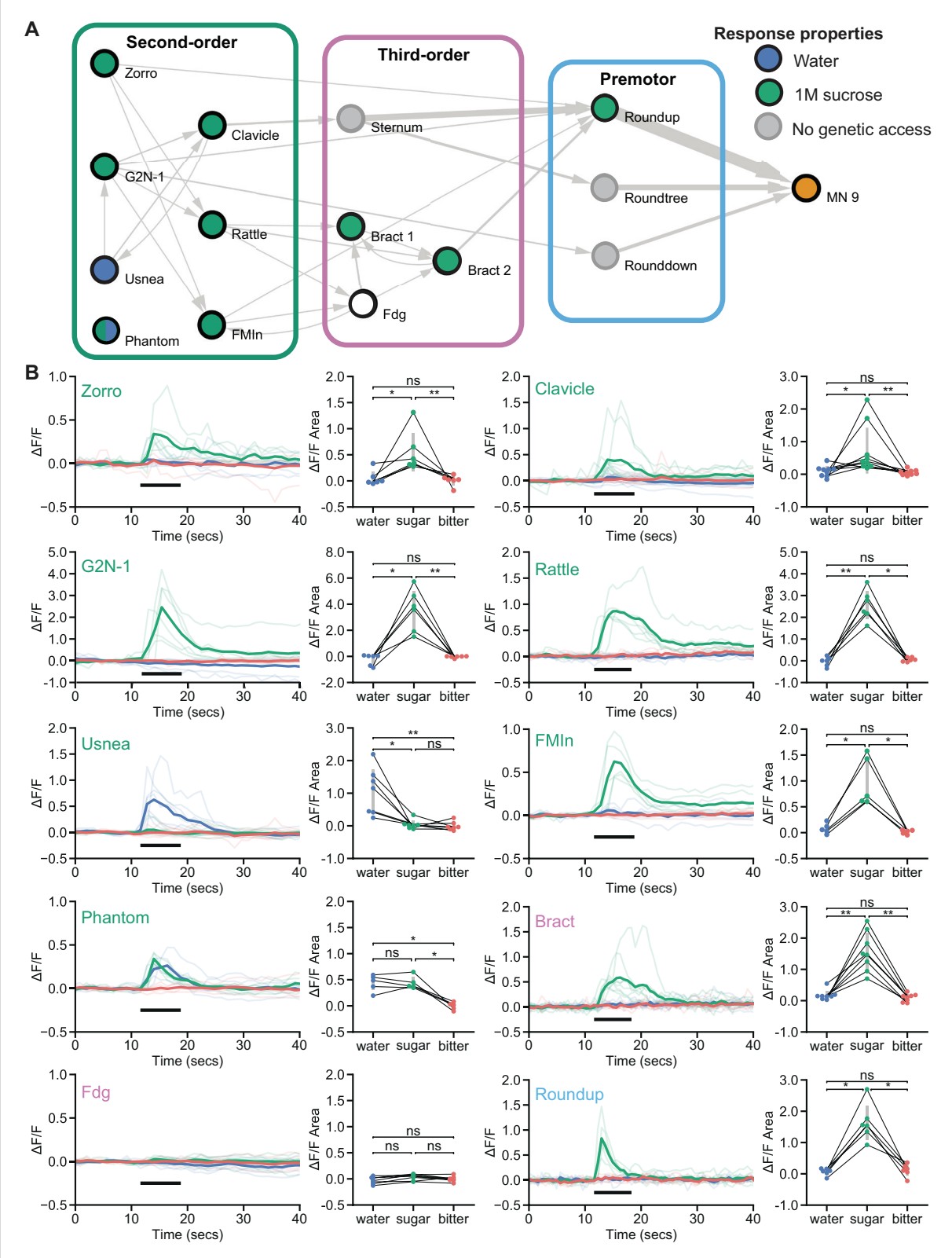

**Figure 3.** Feeding initiation neurons respond to taste detection. (**A**) Connectivity schematic of the feeding initiation circuit, where filled green circles represent cell types that respond to sugar detection, while filled blue circles represent cell types that respond to water detection. One cell type, Phantom, responds to both sugar and water (split blue and green circle). Fdg did not respond to proboscis taste detection (white circle), but see Figure S4A for responses to optogenetic activation of sugar gustatory receptor neurons (GRNs). (**B**) Calcium responses of feeding initiation neurons to

*Figure 3 continued on next page*

*Figure 3 continued*

stimulation of the proboscis in food-deprived flies. For each cell type, GCaMP6s fluorescence traces are shown on the left of the panel (ΔF/F), while ΔF/F area for each trace is shown on the right, with thin black lines indicating sample pairing. The proboscis of each tested individual was stimulated with water (green), sugar (blue), and bitter (red) tastants in sequential trials during the indicated period (thick black line). The following split-GAL4 lines were imaged for each cell type: Clavicle; SS48947, FMIn; SS48944, Zorro; SS67405, G2N-1; SS47082, Usnea; SS37122, Phantom; SS68204, Rattle; SS50091, Fdg; SS31345, Bract; SS31386, Roundup; SS47744. n=5-8 flies per genotype. Quade's test with Quade's All Pairs test, using Holm's correction to adjust for multiple comparisons, ns p>0.05, *p≤0.05, **p≤0.01. See also *Figure 3—figure supplement 1* for additional calcium imaging studies of feeding initiation neurons.

The online version of this article includes the following source data and figure supplement(s) for figure 3:

**Source data 1.** Source data for calcium experiments in *Figure 3*.

**Figure supplement 1.** Taste responses of feeding initiation neurons, related to *Figure 3*.

**Figure supplement 1—source data 1.** Source data for calcium imaging experiments in *Figure 3—figure supplement 1*.

by combining EM tracing studies with precise neural manipulations afforded by split-Gal4 lines, we have elucidated a neural circuit that promotes feeding initiation upon sweet taste detection.

## Feeding initiation neurons respond to sugar taste detection

To examine how taste information is processed by the feeding initiation circuit to guide feeding decisions, we monitored taste-induced activity of each neuron in the circuit. The proboscis was stimulated with water, sugar, or bitter taste solutions, while monitoring GCaMP6s calcium activity (*Chen et al., 2013*) in live flies (*Harris et al., 2015*). Eight of the ten neural classes responded to sugar taste presentation in food-deprived animals, and not to water or bitter solutions (*Figure 3*). Two second-order cell types responded to water taste detection: Usnea responded specifically to water and Phantom responded equally to water and sugar detection. Usnea and Phantom are reciprocally connected with GRNs (*Figure 1—figure supplement 1D*), suggesting that these second-order cell types may tune GRN responses in the presence of water. One third-order neuron, Fdg, did not respond to proboscis taste stimulation, but did respond to optogenetic activation of sugar-sensing GRNs (*Figure 3—figure supplement 1A*), suggesting that Fdg may respond to pharyngeal or leg taste detection. Together, these studies reveal that sugar taste is processed by a multilayered neural circuit to initiate feeding in food-deprived flies.

To test whether responses in the proboscis extension circuit are altered based on specific nutrient needs, we examined taste responses in flies that were thirsty rather than hungry. High hemolymph osmolality is a key signal of thirst that acts on central neurons to promote water consumption (*Jourjine et al., 2016*). We mimicked a thirsty-like state by increasing hemolymph osmolality, which enhanced water responses in water-sensing GRNs (*Figure 3—figure supplement 1B*). In four of the five central neurons tested, response profiles were similar in food-deprived and thirsty-like flies (*Figure 3* and *Figure 3—figure supplement 1C*). However, one second-order neuron, Clavicle, responded to water and to sugar taste detection in a thirsty-like state but only to sugar in a hungry state (*Figure 3* and *Figure 3—figure supplement 1C*). These results suggest that state-dependent responses to water at a single node (Clavicle) may tune the responsiveness of the pathway to bias acceptance of more dilute sugar solutions. However, the majority of neurons uncovered here responded selectively to sugar taste detection regardless of whether the animal was hungry or thirsty.

Given that two second-order neurons, Usnea and Phantom, responded to water taste stimulation despite synaptic connectivity to candidate sugar GRNs, we examined the connections of the second-order feeding initiation neurons from all GRNs of the right hemisphere (*Engert et al., 2021*). These GRNs have previously been clustered into candidate taste categories (sugar, water, bitter, high salt, and low salt) based on their morphology and GRN-GRN connectivity. Remarkably, we found that the second-order neurons that receive sugar GRN inputs also receive inputs from candidate water and high-salt (ppk23-positive, Glut-positive) GRNs but do not receive inputs from candidate bitter or low salt (Ir94e) GRNs (*Figure 3—figure supplement 1D*). The connectivity is consistent with our calcium imaging studies showing responses to sugar taste detection but not bitter taste. However, responses to sugar and water were not consistent with the predicted connectivity for each neuron, suggesting the possibilities of state-dependence, network interactions, and/or errors in GRN modality categorization. As GRN category assignments in the EM dataset were based on anatomy and connectivity alone, some GRNs may be misclassified, leading to errors in assessing sensory inputs (*Engert et al.,*

*2021*). While the connectivity suggests the exciting possibility that taste integration may occur at second-order neurons, further functional studies will be necessary to illuminate the taste categories that activate or inhibit individual second-order neurons under different nutritive states. Nevertheless, our studies demonstrate that most second-order neurons respond to sugar taste stimulation but not to water or bitter tastes.

## Feeding initiation is modulated by hunger at specific nodes

How is sugar taste information integrated with hunger state to promote feeding initiation in food-deprived flies? Hunger modulates sugar GRN activity (*Inagaki et al., 2012*); however, whether sensory gating is the only mechanism for hunger regulation or whether modulation of central neurons contributes to an altered network state in hungry animals has not been examined. To comprehensively investigate how taste detection is integrated with hunger state to initiate feeding, we optogenetically activated each neuron in the PER circuit in either fed or food-deprived flies and examined behavior. Optogenetic activation has the advantage of bypassing changes in sugar sensory detection that propagate through the circuit, enabling the evaluation of central circuit changes.

We reasoned that activating neurons upstream of or at the node(s) where hunger modulation occurs would cause differences in proboscis extension rates between hungry and fed flies, whereas activating neurons beyond the site where hunger impinges would not. Indeed, CsChrimson-mediated activation of sugar GRNs caused higher proboscis extension rates in food-deprived flies than in fed flies, whereas activation of MN9 elicited the same proboscis extension rate in food-deprived and fed flies (*Figure 4A–C*). Moreover, activation of two second-order neurons, G2N-1, and Clavicle, increased proboscis extension in food-deprived flies, whereas activation of all other neural classes did not (*Figure 4A and D*). Thus, hunger signals act on sensory neurons to increase detection sensitivity and on a specific set of second-order interneurons to amplify sugar pathway activation and promote feeding.

## Premotor neurons integrate sweet and bitter taste information

Animals evaluate both internal nutritive state and food quality to decide whether to initiate feeding. To investigate how food quality alters feeding initiation, we examined how the detection of bitter compounds is integrated with sugar taste information in the feeding initiation circuit. Previous studies have demonstrated that bitter compounds inhibit sugar-sensing gustatory neurons to prevent feeding (*Chu et al., 2014*; *French et al., 2015*; *Jeong et al., 2013*; *Meunier et al., 2003*), but have not addressed how downstream neural circuitry modulates appetitive feeding behaviors in response to bitter taste detection. To investigate central mechanisms of bitter modulation, we examined whether pathways from bitter GRNs intersect with the feeding initiation pathway.

As bitter GRNs do not directly synapse with neurons in the feeding initiation circuit, we asked whether second-order bitter neurons synapse onto the feeding initiation pathway. We reconstructed neurons downstream of bitter GRNs (*Engert et al., 2021*) in the EM volume and identified a second-order bitter neuron, Scapula, which receives over 150 synapses from bitter GRNs and is the second-most strongly connected cell type with bitter GRNs (*Figure 5A* and *Figure 5—figure supplement 1*). Scapula synapses directly onto two feeding initiation premotor neurons, Roundup and Rounddown, but not onto second- or third-order appetitive taste neurons.

Because bitter taste detection inhibits proboscis extension, we hypothesized that Roundup and Rounddown would be inhibited by Scapula to prevent proboscis extension to sugar in the presence of bitter compounds. Consistent with this hypothesis, Scapula is predicted to release glutamate (*Figure 2—figure supplement 1D*), which is often an inhibitory neurotransmitter in *Drosophila* (*Liu and Wilson, 2013*). To test this, we monitored activity in Roundup by *in vivo* calcium imaging while activating sugar GRNs, bitter GRNs, or both, using optogenetics to bypass sensory modulation. Roundup responded to optogenetic activation of sugar GRNs but not bitter GRNs (*Figure 5B*), as expected based on its response to taste compounds (*Figure 3B*). Upon co-activation of sugar and bitter GRNs, the Roundup response was dramatically decreased compared to the response to sugar GRN activation alone, arguing that bitter signals suppress the feeding initiation pathway. To test whether this bitter suppression reflects a central mechanism acting at Roundup, we monitored activity in a second-order neuron, G2N-1, directly upstream of Roundup, and found that its response upon co-activation of sugar and bitter GRNs was indistinguishable from its response to sugar GRN

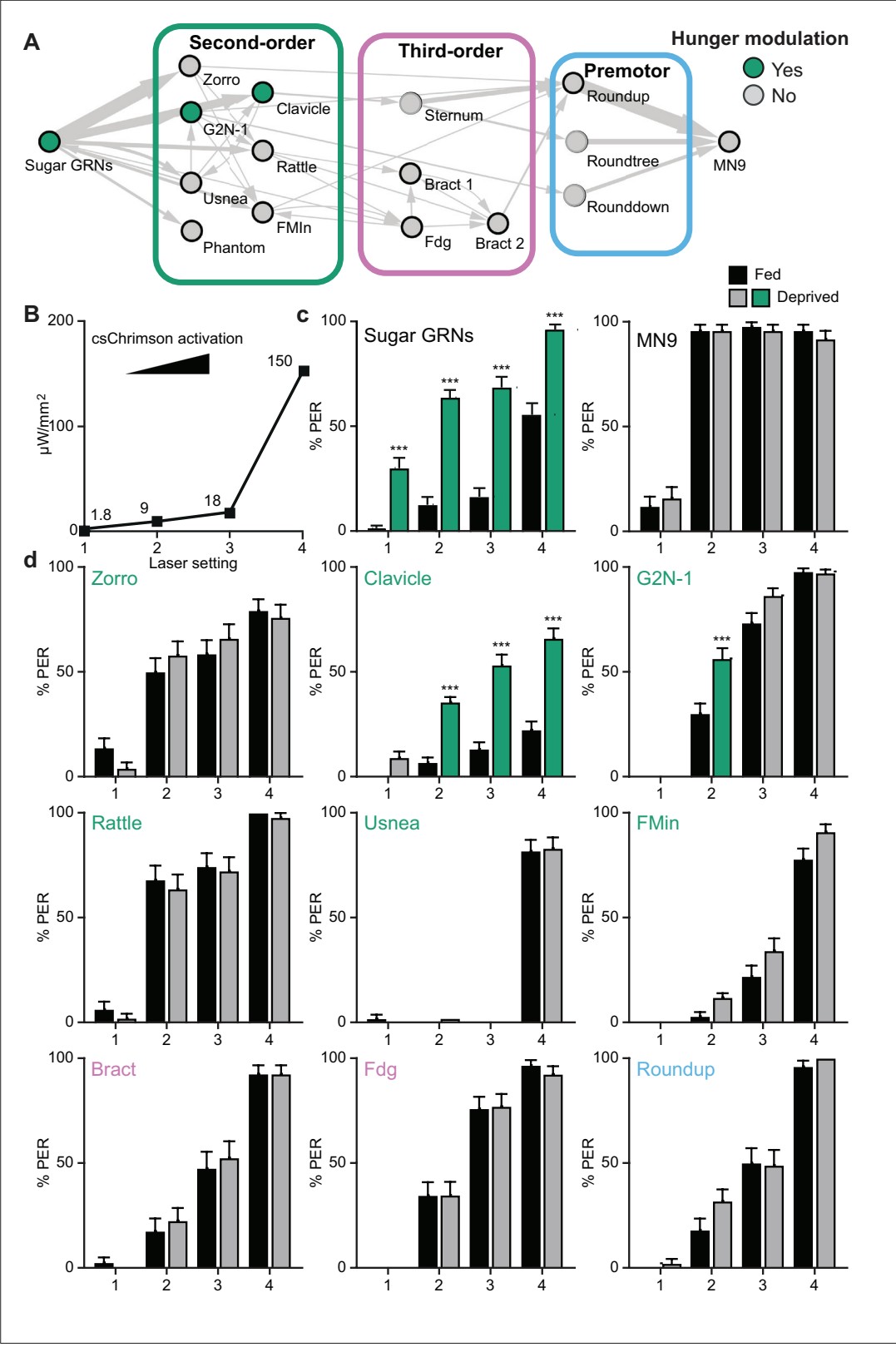

**Figure 4.** Hunger acts on a subset of second-order central neurons to modulate behavior. (**A**) Schematic of the feeding initiation circuit, with filled green circles representing nodes that are hunger-modulated. (**B**) Optogenetic activation at four different light intensities. (**C**) Activation of sugar-sensing neurons results in different feeding initiation rates between fed and food-deprived flies (left) whereas activation of MN9 does not (right), at four

*Figure 4 continued on next page*

*Figure 4 continued*

different light intensities. n=50. (**D**) Optogenetic activation of second-order, third-order, and premotor neurons in either fed or food-deprived flies. n=39–103. Mean ± 95% CI, Fisher's Exact Tests, \*\*\*p<0.001.

The online version of this article includes the following source data for figure 4:

**Source data 1.** Source data for behavioral experiments in *Figure 4*.

activation alone (*Figure 5C*). Together, the EM and imaging studies demonstrate that sugar and bitter tastes are integrated at feeding initiation premotor neurons, providing a central mechanism to reject sweet foods laced with bitter compounds.

## Discussion

In this study, we couple EM circuit reconstruction with the ability to precisely monitor and manipulate single neurons to elucidate how a complex nervous system orchestrates the decision to initiate feeding. First, we delineate the sensorimotor circuit for feeding initiation from sensory inputs to motor outputs with cellular and synaptic resolution. Then, we demonstrate how this central circuit integrates taste detection with internal state, providing mechanistic insight into how taste modalities and feeding decisions are encoded in the brain.

### A local, interconnected network transforms sweet taste detection into behavior

Previous studies in *Drosophila* have identified gustatory neurons, motor neurons, and three candidate interneurons that influence feeding initiation (*Flood et al., 2013*; *Gordon and Scott, 2009*; *Kain and Dahanukar, 2015*; *McKellar et al., 2020*; *Miyazaki et al., 2015*; *Talay et al., 2017*). Here, by elucidating a complete sensorimotor circuit with synaptic resolution, we provide a comprehensive view of the neural pathway that elicits proboscis extension, the first step in feeding. The feeding initiation pathway is a local circuit, with three- and four-synaptic relays to motor output. Each neuron elicits proboscis extension upon optogenetic activation, demonstrating that each neuron participates in a pathway for the behavior. Inhibiting activity of single second-order neurons reduced the behavioral response, whereas inhibiting activity of third-order or premotor neurons did not. As inhibiting activity of single neurons did not abolish proboscis extension, this demonstrates that additional paths not requiring the individual neural cell type contribute to this innate behavior. These results are consistent with the circuit connectivity, which reveals that there are multiple routes between sugar GRNs and MN9 for proboscis extension. The multiple paths from second-order neurons to premotor neurons may enable proboscis extension to be recruited in different contexts to ensure robust feeding. More generally, multiple circuit paths may enhance behavioral flexibility by facilitating sensory tuning and multisensory integration (*Miroschnikow et al., 2018*; *Ohyama et al., 2015*).

Consistent with our finding that there are multiple pathways for proboscis extension, EM circuit tracing of sensorimotor paths for feeding in *Drosophila* larvae revealed both direct GRN-motor neuron synapses as well as indirect pathways (*Miroschnikow et al., 2018*). The functional significance of this circuit organization has not been explored. Although we did not identify direct GRN-motor neuron synaptic connections for proboscis extension in adult *Drosophila*, other feeding subprograms such as pharyngeal pumping may utilize direct sensory to motor connections and remain to be discovered.

Most neurons in the proboscis extension pathway respond to sugar taste detection, but not to water or bitter tastes, in food-deprived flies, demonstrating a direct line from sweet taste detection to the motor output for feeding. How water taste modulates proboscis extension in thirsty flies will require further study. Importantly, we identified and characterized three second-order neurons that respond to water taste detection: one is selective for water taste, another responds to both water and sugar tastes, and the third shows state-dependent sugar and water taste responses. Further study of these second-order neurons and their connectivity will be critical to evaluate the degree of separation or convergence of water and sugar pathways for feeding initiation. Moreover, we find that second-order neurons that respond to sugar receive inputs from sugar GRNs as well as candidate water and high-salt GRNs. This hints at the exciting possibility that second-order neurons integrate multiple taste

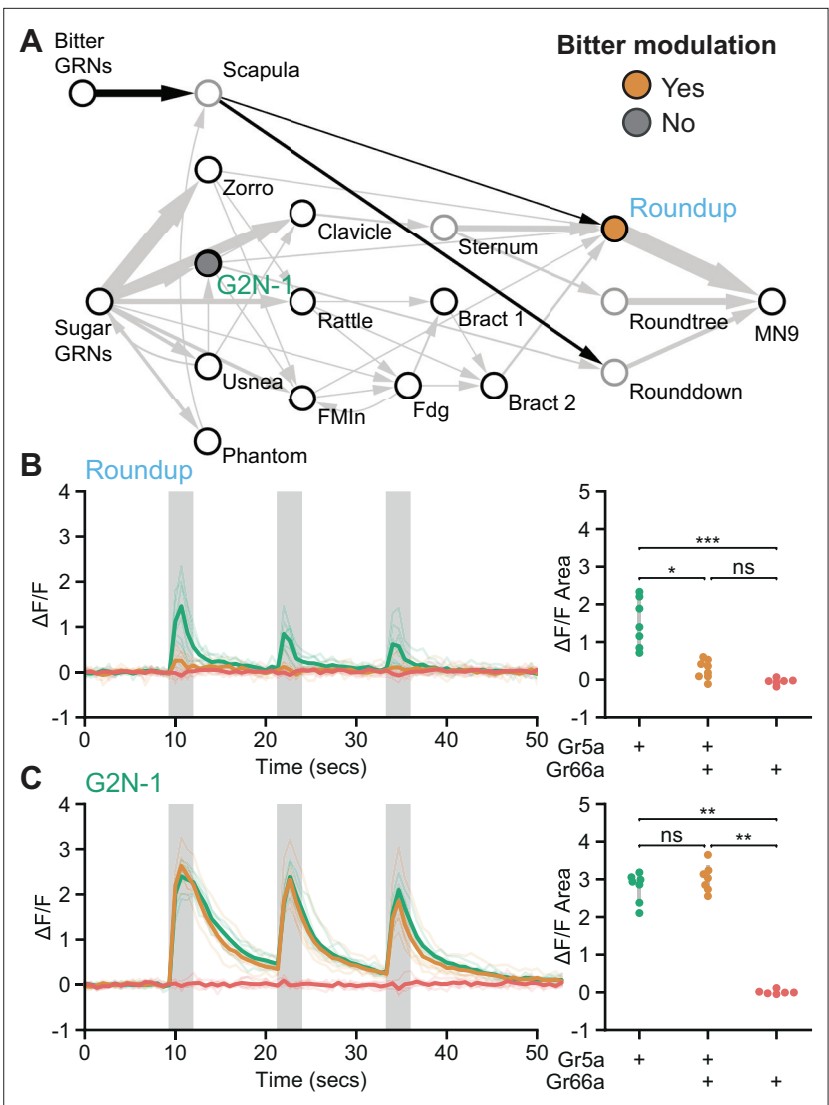

**Figure 5.** Premotor neurons integrate sweet and bitter taste detection. (**A**) Schematic of the feeding initiation circuit, showing a pathway from bitter GRNs to premotor neurons. Filled maize circle labels a premotor neuron inhibited by bitter tastants, filled gray circle labels an upstream second-order neuron that is not inhibited by bitter tastants. (**B and C**) Calcium responses of feeding circuit neurons to optogenetic activation of sugar (green, *Gr5a-LexA*), sugar plus bitter (maize, *Gr5a-LexA* plus *Gr66a-LexA*), or bitter (red, *Gr66a-LexA*) GRNs in food-deprived flies. For each cell type, Syt::GCaMP7b fluorescence traces are shown on the left of the panel (ΔF/F), while ΔF/F area for each trace is shown on the right. Periods of stimulation with 660 nm light are indicated with vertical gray bars. (**B**) SS47744 was imaged to examine Roundup responses. (**C**) SS47082 was imaged to examine G2N-1 responses. (B-C) Kruskal Wallace test with Dunn's test using Holm's correction to adjust for multiple comparisons, n=6-8 flies per genotype, ns p>0.05, *p≤0.05, **p≤0.01, ***p≤0.001. See **Figure 5—figure supplement 1** for synaptic counts of second-order bitter neurons.

The online version of this article includes the following source data and figure supplement(s) for figure 5:

**Source data 1.** Source data for calcium imaging experiments in **Figure 5**.

**Figure supplement 1.** Second-order bitter neurons, related to **Figure 5**.

---

modalities and may show responses that vary based on nutritive state. Furthermore, functional studies will be essential to assess taste integration by second-order neurons.

Of the interneurons identified here, only G2N-1 and Fdg have previously been implicated in feeding. G2N-1 was identified as a candidate sugar-sensing second-order neuron based on its anatomical proximity to gustatory axons alone (**Miyazaki et al., 2015**); here, we elucidated its functional role in

taste detection and feeding initiation. Fdg was isolated as a 'feeding command neuron,' able to elicit multiple steps in feeding, including proboscis extension (*Flood et al., 2013*). In our calcium imaging studies, Fdg did not respond to proboscis taste stimulation but did respond to optogenetic activation of sugar GRNs. This suggests that Fdg may receive gustatory signals from GRNs on the pharynx or legs. Our studies demonstrate that Fdg is a third-order neuron in the feeding initiation pathway, with synaptic connections to Bract descending neurons. A description of the reconstruction of all FlyWire DNs is in preparation (K. Eichler, M. Costa, G. Card, G. Jefferis, personal communication). As Bract synapses with proboscis premotor neurons and ventral cord circuits, Fdg and Bract are well-poised to coordinate proboscis extension with other steps in feeding.

The architecture of the circuit provides a platform to investigate how taste signals are transformed in the brain to drive behavior. In this study, we focused on MN9, the rostrum protractor motor neuron that elicits proboscis extension, as a key readout of proboscis extension behavior. However, proboscis extension involves not only rostrum protraction but also extension of the haustellum and opening of the labellum, controlled by additional motor neurons (*McKellar et al., 2020*). We hypothesize that the connectivity among second- and third-order neurons may coordinate the precise temporal activation of different muscle groups for coordinated extension. Moreover, proboscis extension is followed by ingestion and then meal termination (*Dethier, 1976*; *Pool and Scott, 2014*). Continued expansion and exploration of this pathway will provide the opportunity to examine how different feeding subprograms are timed and coordinated to elicit feeding in natural environments.

## Hunger tunes second-order neurons to promote sugar responses

Studies in *C. elegans*, *Drosophila*, and mammals have demonstrated that a key site of hunger regulation is at the peripheral chemosensory neurons, altering sensitivity of detection (*Chalasani et al., 2010*; *Kawai et al., 2000*; *Root et al., 2011*; *Savigner et al., 2009*; *Sengupta, 2013*). For example, dopamine enhances the sensitivity of *Drosophila* sugar-sensing gustatory neurons to promote proboscis extension at lower sucrose concentrations in hungry animals (*Inagaki et al., 2012*; *Marella et al., 2012*). Hunger modulation of taste processing beyond sensory neurons has been more challenging to evaluate, both because of lack of knowledge of central networks and because changes at the sensory level propagate through the network.

To isolate the role of central brain neurons in hunger modulation, we used the precise genetic access available in *Drosophila* to activate each node of the feeding initiation pathway and examined the behavioral response elicited in fed and food-deprived flies. These studies pinpoint the site of hunger modulation to sensory neurons and two second-order neurons. Although caveats of artificial stimulation exist, the consistent changes seen across different light intensities for neural manipulation early in the pathway, but not downstream, argue that these results are robust. These studies demonstrate that hunger acts at a few critical nodes to modulate feeding initiation: sensory neurons increase detection sensitivity and second-order neurons amplify pathway activation. It will be interesting to examine whether hunger modulation of sensory and second-order neurons occurs independently or over different time scales to adjust behavioral responses as starvation increases. In addition, the specific hunger signals that act on central neurons and their mechanism of modulation may now be explored.

## Bitter compounds inhibit premotor neurons to prevent feeding initiation

While previous studies have demonstrated interactions between sweet and bitter taste modalities at the level of sensory neurons (*Chu et al., 2014*; *French et al., 2015*; *Jeong et al., 2013*; *Meunier et al., 2003 LeDue et al., 2016*) and through feedback from the mammalian gustatory cortex (*Jin et al., 2021*), this study reveals a third circuit strategy for weighing sweet and bitter tastes: a local inhibitory network. Inhibitory interactions between bitter and sugar pathways at the level of premotor neurons provide an elegant strategy to weigh incoming sugar and bitter taste information and adjust behavioral probability. In addition, by blocking activity at specific muscles, bitter detection may specifically change behavior to direct the proboscis away from a hazardous food source. The existence of a local inhibitory circuit for bitter-sweet integration has been recently postulated based on studies of mammalian taste circuitry (*Jin et al., 2021*) and may be a shared strategy across species. These

multiple circuit mechanisms for suppression of sweet attraction by bitter signals may reflect the evolutionary importance of robust bitter taste avoidance.

By examining a complete sensorimotor pathway, we elucidate how a complex nervous system orchestrates the decision to initiate feeding and illuminate central modules that integrate taste detection with internal state. These central controls afford independent amplification and suppression of feeding and stand apart from sensory modulation as mechanisms that dynamically tune behavior. As sensory modulation may suffer from finite amplification and incomplete suppression, central modulation provides a strategy to bypass those limits, allowing a broader range and different temporal dynamics of modulation.

# Materials and methods

**Key resources table**

| Reagent type (species) or resource | Designation | Source or reference | Identifiers | Additional information |
|---|---|---|---|---|
| Antibody | Anti-Brp (mouse monoclonal) | DSHB, University of Iowa, USA | DSHB Cat# nc82, RRID:AB_2314866 | 1/500 |
| Antibody | Anti-GFP (chicken polyclonal) | Thermo Fisher Scientific | Thermo Fisher Scientific Cat# A10262, RRID:AB_2534023 | 1/1000 |
| Antibody | Anti-dsRed (rabbit polyclonal) | Takara | Takara Bio Cat# 632496, RRID:AB_10013483 | 1/1000 |
| Antibody | Anti-chicken Alexa Fluor 488 (goat polyclonal) | Thermo Fisher Scientific | Thermo Fisher Scientific Cat# A-11039, RRID:AB_2534096 | 1/100 |
| Antibody | Anti-rabbit Alexa Fluor 568 (goat polyclonal) | Thermo Fisher Scientific | Thermo Fisher Scientific Cat# A-11036, RRID:AB_10563566 | 1/100 |
| Antibody | Anti-mouse Alexa Fluor 647 (goat polyclonal) | Thermo Fisher Scientific | Thermo Fisher Scientific Cat# A-21236, RRID:AB_2535805 | 1/100 |
| Chemical Compound, drug | All trans-Retinal | MilliporeSigma | Cat # R2500 | |
| Genetic reagent (*D. melanogaster*) | 20XUAS-IVS-CsChrimson.mVenus attP18 | Bloomington Stock Center; *Klapoetke et al., 2014* | RRID:BDSC_55134 | |
| Genetic reagent (*D. melanogaster*) | UAS-GtACR1.d.EYFP}attP2 | Bloomington Stock Center | RRID:BDSC_92983 | |
| Genetic reagent (*D. melanogaster*) | Zorro split-GAL4, SS67405 | Janelia Research Campus | | Full genotype: w; R12C04-p65ADZp in attP40; VT043788-ZpGDBD in attP2 |
| Genetic reagent (*D. melanogaster*) | Zorro split-GAL4, SS67406 | Janelia Research Campus | | Full genotype: w; R12C04-p65ADZp in attP40; VT020600-ZpGDBD in attP2 |
| Genetic reagent (*D. melanogaster*) | Clavicle split-GAL4, SS48947 | Janelia Research Campus; *Sterne et al., 2021*; available at http://splitgal4.janelia.org | | Full genotype: w; VT020732-p65ADZp in attP40; R17G10-ZpGDBD in attP2 |
| Genetic reagent (*D. melanogaster*) | G2N-1 split-GAL4, SS47082 | Janelia Research Campus; *Sterne et al., 2021*; available at http://splitgal4.janelia.org | | Full genotype: w; R12C04-p65ADZp in attP40; VT043658-ZpGDBD in attP2 |

*Continued on next page*

*Continued*

| Reagent type (species) or resource | Designation | Source or reference | Identifiers | Additional information |
|---|---|---|---|---|
| Genetic reagent (*D. melanogaster*) | G2N-1 split-GAL4, SS56399 | Janelia Research Campus; *Sterne et al., 2021*; available at http://splitgal4.janelia.org | | Full genotype: w; R12C04-p65ADZp in attP40; VT020839-ZpGDBD in attP2 |
| Genetic reagent (*D. melanogaster*) | Rattle split-GAL4, SS50091 | Janelia Research Campus; *Sterne et al., 2021*; available at http://splitgal4.janelia.org | | Full genotype: w; VT006545-p65ADZp in attP40; VT023745-ZpGDBD in attP2 |
| Genetic reagent (*D. melanogaster*) | Usnea split-Gal4, SS37122 | Janelia Research Campus; *Sterne et al., 2021*; available at http://splitgal4.janelia.org | | Full genotype: w; VT037525-p65ADZp in attP40; VT033627-ZpGDBD in attP2 |
| Genetic reagent (*D. melanogaster*) | Usnea and Cleaver split-Gal4, SS31022 | Janelia Research Campus; *Sterne et al., 2021*; available at http://splitgal4.janelia.org | | Full genotype: w; VT038544-p65ADZp in attP40; VT019345-ZpGDBD in attP2 |
| Genetic reagent (*D. melanogaster*) | FMIn split-GAL4, SS48944 | Janelia Research Campus; *Sterne et al., 2021*; available at http://splitgal4.janelia.org | | Full genotype: w; R81E10-p65ADZp in attP40; R17G10-ZpGDBD in attP2 |
| Genetic reagent (*D. melanogaster*) | FMIn split-GAL4, SS48949 | Janelia Research Campus; *Sterne et al., 2021*; available at http://splitgal4.janelia.org | | Full genotype: w; R81E10-p65ADZp in attP40; R21H11-ZpGdbd in attP2 |
| Genetic reagent (*D. melanogaster*) | Phantom split-GAL4, SS43877 | Janelia Research Campus; *Sterne et al., 2021*; available at http://splitgal4.janelia.org | | Full genotype: w; R82F02-p65ADZp in attP40; R20G06-ZpGdbd in attP2 |
| Genetic reagent (*D. melanogaster*) | Phantom split-GAL4, SS43879 | Janelia Research Campus; *Sterne et al., 2021*; available at http://splitgal4.janelia.org | | Full genotype: w; R20G06-p65ADZp in attP40; R82F02-ZpGDBD in attP2 |
| Genetic reagent (*D. melanogaster*) | Phantom split-GAL4, SS68204 | Janelia Research Campus | | Full genotype: w; R20G06-p65ADZp in attP40; R81A07-ZpGdbd in attP2 |
| Genetic reagent (*D. melanogaster*) | Fudog split-GAL4, SS35290 | Janelia Research Campus; *Sterne et al., 2021*; available at http://splitgal4.janelia.org | | Full genotype: w; R59F08-p65ADZp in attP40; R69E06-ZpGDBD in attP2 |
| Genetic reagent (*D. melanogaster*) | Fudog split-GAL4, SS35291 | Janelia Research Campus; *Sterne et al., 2021*); available at http://splitgal4.janelia.org | | Full genotype: w; VT038225-p65ADZp in attP40; R69E06-ZpGDBD in attP2 |
| Genetic reagent (*D. melanogaster*) | Bract split-GAL4, SS31320 | Janelia Research Campus; *Sterne et al., 2021*; available at http://splitgal4.janelia.org | | Full genotype: w; R25A01-p65ADZp in attP40; VT058723-ZpGDBD in attP2 |
| Genetic reagent (*D. melanogaster*) | Bract split-GAL4, SS31386 | Janelia Research Campus; *Sterne et al., 2021*; available at http://splitgal4.janelia.org | | Full genotype: w; R25A01-p65ADZp in attP40; R37D11-ZpGDBD in attP2 |
| Genetic reagent (*D. melanogaster*) | Fdg split-GAL4, SS31333 | Janelia Research Campus; *Sterne et al., 2021*; available at http://splitgal4.janelia.org | | Full genotype: w; R81E10-p65ADZp in attP40; VT037804-ZpGDBD in attP2 |
| Genetic reagent (*D. melanogaster*) | Fdg split-GAL4, SS46913 | Janelia Research Campus; *Sterne et al., 2021*; available at http://splitgal4.janelia.org | | Full genotype: w; R81E10-p65ADZp in attP40; R88C07-ZpGdbd in attP2 |

*Continued*

| Reagent type (species) or resource | Designation | Source or reference | Identifiers | Additional information |
|---|---|---|---|---|
| Genetic reagent (*D. melanogaster*) | Roundup split-GAL4, SS47744 | Janelia Research Campus; *Sterne et al., 2021*; available at http://splitgal4.janelia.org | | Full genotype: w; R23G11-p65ADZp in attP40; VT003236-ZpGDBD in attP2 |
| Genetic reagent (*D. melanogaster*) | Roundup split-GAL4, SS47745 | Janelia Research Campus; *Sterne et al., 2021*; available at http://splitgal4.janelia.org | | Full genotype: w; R11B11-p65ADZp in attP40; VT003236-ZpGDBD in attP2 |
| Genetic reagent (*D. melanogaster*) | 20xUAS >dsFRT > csChrimson-mVenus | *Wu et al., 2016* | | |
| Genetic reagent (*D. melanogaster*) | 8XLexAop2-FLPL(attP40) | Bloomington Stock Center | RRID:BDSC_55820 | |
| Genetic reagent (*D. melanogaster*) | ;;Dfd-LexA | *Simpson, 2016* | | |
| Genetic reagent (*D. melanogaster*) | ;;Scr-LexA | *Simpson, 2016* | | |
| Genetic reagent (*D. melanogaster*) | w[1118]; 20XUAS-IVS-GCaMP6s(attP40); | Bloomington Stock Center | RRID: BDSC_42746 | |
| Genetic reagent (*D. melanogaster*) | Gr64f-Gal4 (II) | *Kwon et al., 2011* | | |
| Genetic reagent (*D. melanogaster*) | Ppk28-Gal4 | Bloomington Stock Center; *Cameron et al., 2010* | RRID:BDSC_93020 | |
| Genetic reagent (*D. melanogaster*) | Gr64f-LexA | *Miyamoto et al., 2012* | | |
| Genetic reagent (*D. melanogaster*) | Gr66a-LexA(II) | *Thistle et al., 2012*; Bloomington Stock Center; | RRID:BDSC_93023 | |
| Genetic reagent (*D. melanogaster*) | Gr66a-LexA5(III) | *Thistle et al., 2012* | | |
| Genetic reagent (*D. melanogaster*) | UAS-CD8-tdTomato;; | *Thistle et al., 2012* | | |
| Genetic reagent (*D. melanogaster*) | w[1118]; 20XUAS-IVS-GCaMP6s(attP40); | Bloomington *Drosophila* Stock Center | RRID: BDSC_42746 | |
| Genetic reagent (*D. melanogaster*) | w[1118];; 20XUAS-IVSGCaMP6s(VK00005) | Bloomington *Drosophila* Stock Center | RRID: BDSC_42749 | |
| Genetic reagent (*D. melanogaster*) | Gr5a-LexA-VP16(II) | *Gordon and Scott, 2009* | RRID:BDSC_93014 | |
| Genetic reagent (*D. melanogaster*) | ppk28-LexA(III) | *Thistle et al., 2012* | | |
| Genetic reagent (*D. melanogaster*) | 13XLexAop2-IVS-p10-ChrimsonR-mCherry(attP18) | Vivek Jarayaman | | |
| Genetic reagent (*D. melanogaster*) | 20XUAS-IVS-jGCaMP7b(attP5) | Bloomington Stock Center | RRID:BDSC_80907 | |
| Genetic reagent (*D. melanogaster*) | 20XUAS-IVS-jGCaMP7b(VK00005) | Bloomington Stock Center | RRID:BDSC_79029 | |
| Genetic reagent (*D. melanogaster*) | 20xUAS-IVS-Syn21-Syt::Op-jGCaMP7b(attP18) | Vivek Jarayaman, Chuntao Dan | | |
| Software, Algorithm | Fiji | https://fiji.sc/ | RRID: SCR_002285 | |
| Software, Algorithm | Computational Morphometry Toolkit (CMTK) | *Masse et al., 2012* | | |

*Continued on next page*

*Continued*

| Reagent type (species) or resource | Designation | Source or reference | Identifiers | Additional information |
|---|---|---|---|---|
| Software, Algorithm | NBLAST | *Costa et al., 2016*; http://nblast.virtualflybrain.org:8080/NBLAST_on-the-fly/; http://flybrain.mrc-lmb.cam.ac.uk/si/nblast/www/ | | |
| Software, Algorithm | VVDviewer | *Otsuna et al., 2018*; https://github.com/takashi310/VVD_Viewr | | |
| Software, Algorithm | GraphPad Prism | Graphpad Software; https://www.graphpad.com/scientific-software/prism/ | RRID:SCR_002798 | |
| Software, Algorithm | Python | Python Software Foundation; https://www.python.org/downloads/ | | |
| Software, Algorithm | Flywire | Flywire; https://flywire.ai/ | RRID:SCR_019205 | |
| Software, Algorithm | Adobe Illustrator | Adobe Software; https://www.adobe.com/products/illustrator.html | | |
| Software, Algorithm | CATMAID | *Saalfeld et al., 2009*; https://catmaid.org | | |
| Software, Algorithm | CAVE (connectome annotation versioning engine) | https://github.com/seung-lab/CAVEclient/blob/master/FlyWireSynapseTutorial.ipynb | | |
| Software, Algorithm | R Project for Statistical Computing | *R Development Core Team, 2018* | RRID:SCR_001905 | |
| Software, Algorithm | CircuitCatcher | *Bushey, 2019*; https://github.com/DanBushey/CircuitCatcher | | |
| Software, Algorithm | PMCMRplus package | *Pohlert, 2021*; https://CRAN.R-project.org/package=PMCMRplus | | |
| Software, Algorithm | SciPy package | *Virtanen et al., 2020*; https://scipy.org/ | | |
| Software, Algorithm | scikit-posthocs package | *Terpilowski, 2018*; https://scikit-posthocs.readthedocs.io/en/latest/ | | |

## Experimental model and subject details

### Rearing conditions and strains

All experiments were performed in the fruit fly *Drosophila melanogaster*. The key resources table lists the transgenic lines used in this study. Flies were reared on standard cornmeal-yeast-molasses media at 25°C with 65% humidity and a 12 hr: 12 hr light: dark cycle unless stated otherwise. Flies for optogenetic experiments were raised on standard food in darkness. Upon eclosion, adult flies were collected and maintained on standard food supplemented with 0.4 mM all-trans-retinal in darkness prior to experiments. Adult mated female flies were used for all experiments.

## Method details

### EM neural reconstructions

Neurons were reconstructed in a serial section transmission electron volume (Full Adult Female Brain, *Zheng et al., 2018*) using the CATMAID software (*Saalfeld et al., 2009*). Fully manual reconstructions were generated by following the branches of the neuron and marking the center of each branch, thereby creating a 'skeleton' of each neuron. In addition to fully manual reconstructions, segments of an automated segmentation (*Li et al., 2019*) were proofread and expanded to generate complete reconstructions.

Seventeen candidate sugar GRNs in the right hemisphere were previously identified in the EM connectome by clustering GRNs using morphology and connectivity data and comparing the resulting clusters with immunostained GRNs responding to different taste categories (*Engert et al., 2021*). We specifically reconstructed second-order sugar neurons downstream of the candidate sugar GRNs in the right hemisphere using two different methods. First, random presynapses of skeleton 7349219

(*Engert et al., 2021*) were chosen using the reconstruction sampler function of CATMAID and downstream partners were reconstructed. Second, large automatically generated fragments downstream of sugar GRN axons were found and expanded. Chemical synapses were annotated as previously described (*Zheng et al., 2018*); specifically, at least three of four elements of a synapse were needed to call a synapse: a T-bar, postsynaptic density, synaptic vesicles, and a synaptic cleft. All reconstructions for which there is a corresponding split-Gal4 were assembled and proofread to near completion.

## Neuron nomenclature

The vast majority of the neurons referred to here were named in *Sterne et al., 2021*. Zorro was named because the proximal neurite forms a 'Z.' Scapula was named due to its resemblance to an inverted scapula bone, and Sternum was named due to its appearance, connectivity, and proximity to Clavicle. Roundtree and Rounddown were named because they, like Roundup (named in *Sterne et al., 2021*), are premotor neurons.

## Flywire connectivity analysis

Neurons corresponding to those traced in CATMAID were located in Flywire (Flywire.ai); both reconstructions use the same underlying EM data (*Zheng et al., 2018*). To identify neurons upstream or downstream of a set of Flywire neurons, we used CAVE (connectome annotation versioning engine; *Buhmann et al., 2021*; *Heinrich et al., 2018*). To identify synapses of fairly high confidence, we chose a 'cleft_score' cutoff of 100 (*Heinrich et al., 2018*).

The CATMAID skeleton IDs and Flywire IDs for each reconstructed neurons are listed here: Billiards (CATMAID: 8606542, Flywire: 720575940634231886), Bract1 (17024882, 720575940625204508), Bract2 (17542353, 720575940637873717), Clavicle (10150139, 720575940620111024), Dandelion (17249809, 720575940628601052), Fdg (16783943, 720575940632291554), FMIn (8952676, 720575940645551748), Fuchs (7929209, 720575940623691196), Fudog (7983275, 720575940630459463), G2N-1 (15079937, 720575940606258268), MN9 (16866694, 720575940616055252), Phantom (16762541, 720575940618879604), Quasimodo (8275570, 720575940619419814), Rattle (16238926, 720575940608777796), Rounddown (16886973, 720575940609112018), Roundup (16002203, 720575940620364549), Scapula (16887116, 720575940624539966), Specter (17579359, 720575940616547141), Sternum (17533840, 720575940643288356), Usnea (14890522, 720575940615947993), Zorro L (7574284, 720575940643219566), and Zorro R (7899212, 720575940629888530). FAFB neuronal reconstructions will be available from Virtual Fly Brain (https://fafb.catmaid.virtualflybrain.org/).

## Genetic access to Cleaver

To gain specific genetic access to Cleaver, we used a triple intersection approach. In this approach, CsChrimson-mVenus will only be expressed where the expression patterns of the AD, DBD, and LexA overlap. SS31022 (*Sterne et al., 2021*) labels both Cleaver and Usnea. To specifically access Cleaver, virgins of 20xUAS >dsFRT > csChrimson-mVenus;8XLexAop2-FLPL(attP40);Dfd-LexA were crossed to males of SS31022. To specifically access Usnea in SS31022, virgins of 20xUAS >dsFRT > csChrimson-mVenus;8XLexAop2-FLPL(attP40);Scr-LexA were crossed to males of SS31022. For each intersection, female progeny without balancers were selected for behavioral analysis.

To visualize triple intersection expression patterns, brains were dissected as described (https://www.janelia.org/project-team/flylight/protocols, 'Dissection and Fixation 1.2% PFA').

The following primary antibodies were used:

- –1:40 mouse α-Brp (nc82) (DSHB, University of Iowa, USA).
- –1:1000 chicken α-GFP (Invitrogen A10262).

The following secondary antibodies were used:

- –1:500 α-mouse AF647 (Invitrogen, A21236).
- –1:1000 α-chicken AF488 (Life Technologies, A11039).

Immunohistochemistry was carried out as described (https://www.janelia.org/project-team/flylight/protocols, 'IHC-Anti-GFP') substituting the above antibodies and eschewing the pre-embedding fixation steps. Ethanol dehydration and DPX mounting was carried out as described (https://www.janelia.

org/project-team/flylight/protocols, 'DPX Mounting'). Images were acquired with a Zeiss LSM 880 NLO AxioExaminer at the Berkeley Molecular Imaging Center. A Plan-Apochromat 25×/0.8 objective was used at zoom 0.7. Acquired images had a voxel size of 0.59 μm × 0.59 μm × 1.50 μm.

## Identification of split-GAL4 lines from EM reconstructions

NBLAST analysis was used to match neurons reconstructed in EM to neurons labeled by split-GAL4 lines (*Costa et al., 2016*). Reconstructed neurons from CATMAID were transformed into the JRC2018U template space using NAVIS (*Bates et al., 2020a*; *Schlegel et al., 2021*) and compared to a light-level library of 122 SEZ cell types in the SEZ split-GAL4 collection (*Sterne et al., 2021*). In addition, we added a representative image from a split-GAL4 we designed to cover a cell type reported here, Zorro, using previously described methods (*Sterne et al., 2021*). Each reconstructed neuron on the right of the brain was compared to every SEZ cell type in the library using the natverse toolkit in R (*Bates et al., 2020b*). Normalized, mean scores were calculated to control for neuron size and segment number. The highest scoring light-level cell type for each reconstructed neuron was considered a match if the normalized, mean NBLAST score was greater than 0.4.

Reconstructed cell types with matches include the following *FAFB IDs* (Top match cell type, NBLAST score): *Bract1* (Bract, 0.58), *Bract2* (Bract, 0.57), *Clavicle* (Clavicle, 0.54), *Cleaver* (Cleaver, 0.57), *Fdg* (Fdg, 0.64), *FMIn* (FMIn, 0.61), *Fudog R* (Fudog, 0.43), *G2N-1* (G2N-1, 0.49), *Phantom* (Phantom, 0.67), *Rattle* (Rattle, 0.60), *Roundup* (0.63), *Usnea* (Usnea, 0.50), *Zorro R* (Zorro, 0.54).

Reconstructed cell types which did not return matches include the following *FAFB IDs* (Top match cell type, NBLAST score): *Billiards* (Phantom, 0.17), *Buster* (Marge, 0.30), *Dandelion* (Clavicle, –0.16), *Fuchs* (Phantom, 0.37), *Quasimodo* (Puddle, 0.32), *Rounddown* (Roundup, 0.28), *Roundtree* (Puddle, 0.27), *Specter* (Usnea, 0.16), *Sternum* (Rattle, 0.10).

Flywire dense reconstructions of second- and third-order neurons (*Figure 1—figure supplement 1F*; *Figure 2—figure supplement 1B*) identified no anatomically indistinguishable neurons, except for Bract 1 and Bract 2. Therefore, the high similarity in projection patterns between the split-GAL4 lines and the EM neurons, as well as the functional and behavioral responses of the split-GAL4 lines, provide confidence that the neurons labeled by the split-Gal4 lines match the EM neurons.

## Optogenetic activation

PER was scored as previously described (*Mann et al., 2013*). Female flies were raised on standard cornmeal-yeast-molasses medium, until 48 hr before experiments, when flies were placed on molasses food with 0.4 mM retinal. Three- to five-day-old flies were anesthetized with carbon dioxide, mounted onto a glass slide with nail polish, and allowed to recover for 2 hr in a humidified chamber at 22°C. For optogenetic activation experiments, 153 uW/mm$^2$ 635 nm laser light was used (Laserglow). Flies were scored for whether they extended their proboscis within a 5 s period in response to light. Experiments were performed blind to genotype.

For food-deprivation experiments, flies were raised as above, except 48 hr before experiments, flies were wet-starved by placing them in a vial with a water saturated kimwipe supplemented with 0.4 mM retinal. Flies were activated with a 635 nm laser at four different light intensities: 1.8, 8.9, 17.8, and 153 uW/mm$^2$.

## GtACR1 silencing

Three-day-old female flies were raised on standard food, and transferred to standard food with 0.4 mM all-trans retinal for 2 days. Next, flies were wet-starved with 0.4 mM retinal in water for 24 hr in order to identify decreases in proboscis extension. Flies were anesthetized with carbon dioxide, mounted onto a glass slide with nail polish, and allowed to recover for 2 hr in a humidified chamber at 22°C. A green laser (532 nm, LaserGlow LBS-532) was used to acutely silence neurons using GtACR1 (*Mohammad et al., 2017*). Flies were water satiated, then presented with either 50 mM sucrose or 100 mM sucrose three times to the proboscis, and the number of flies that extended at least once was recorded.

## *In vivo* sample preparation for calcium imaging

Mated female flies were dissected for calcium imaging studies 14–21 days post-eclosion as previously described (*Harris et al., 2015*) with the following modifications. Flies were briefly anesthetized with ice as they were placed in a custom plastic holder at the cervix to isolate the head from the rest of the body. The head was then immobilized using UV glue, and the esophagus was cut to provide unobstructed imaging access to the SEZ. Flies in fed, food-deprived, desiccated, and thirsty-like (pseudodessicated) conditions were generated as follows:

Fed: Flies were placed in a fresh vial containing standard cornmeal-yeast-molasses media 18–24 hr prior to imaging. Following dissection, samples were bathed in ~250 mOsmo Artificial Hemolymph-Like solution (AHL) ('artificial hemolymph') and imaged immediately.

Food-deprived: Flies were food-deprived in a vial containing a wet kimwipe for 18–24 hr prior to imaging. Following dissection, samples were bathed in ~250 mOsmo AHL and imaged immediately.

Desiccated: Flies were placed in a vial containing 5 grams of Drierite for 2 hr. A cotton ball was used to isolate flies from the desiccant inside the vial, and the vial was closed with parafilm to create a dry chamber. Following dissection, samples were bathed in ~250 mOsmo AHL and imaged immediately. Hemolymph signals of thirst, such as osmolality, may be perturbed in our calcium imaging studies, limiting our ability to accurately assess a thirsty state (*Jourjine et al., 2016*).

Thirsty-like (Pseudodessicated): Flies were placed in a fresh vial containing standard cornmeal-yeast-molasses media 18–24 hr prior to imaging. Following dissection, samples were bathed in ~350 mOsmo AHL ('high osmolality artificial hemolymph') and allowed to rest for 1 hr prior to imaging.

## Calcium imaging with taste stimulation

For imaging responses to taste solutions, females of UAS-CD8-tdTomato;20XUAS-IVS-GCaMP6s(attP5);20XUAS-IVS-GCaMP6s(VK00005) were crossed to males for each split-GAL4 line, and female progeny without balancers were selected for imaging. We found that the arborizations of single neurons were easier to locate *in vivo* when two copies of GCaMP6s were used, likely due to weaker GAL4 expression in the split-GAL4 lines. The following tastants were used: double-distilled water ('water'), 1 M sucrose ('sugar'), or 10 mM denatonium plus 100 mM caffeine in 20% polyethylene glycol (PEG) ('bitter'). Taste solutions were delivered to the proboscis using a glass capillary (1.0 mm OD/ 0.78 mm ID) filled with ~4 µL of taste solution and positioned at the tip of the proboscis using a micromanipulator. Taste solutions were drawn away from the tip of the capillary at the beginning of each imaging trial using slight suction generated by an attached 1 mL syringe, and delivered to the proboscis at the relevant time during imaging with light pressure applied to the syringe.

Calcium imaging was performed using either a 1- or 2-photon microscope. For cell types in close proximity to the surface of the SEZ, 1-photon imaging was performed using a 3i spinning disc confocal microscope with a piezo drive and a 20 × water immersion objective (NA = 1.0) with a 2.5 × magnification changer. 55 frames of 8 z sections spaced at 1 µM intervals were binned 4 × 4 and acquired at 0.8 Hz using a 488 nm laser. Taste solutions were in contact with the proboscis labellum from frame 20 to frame 25. Cell types imaged using a 1-photon microscope are Clavicle, Fdg, FMIn, G2N-1, Phantom, Usnea, and Zorro. For cell types that arborize deeper in the SEZ, 2-photon imaging was performed using a Scientifica Hyperscope with resonant scanning, a piezo drive, and a 20× water immersion objective (NA = 1.0) with 4× digital zoom. 80 stacks of 20 z sections spaced at 2 µM intervals were acquired at 0.667 Hz using a 920 nm laser. Taste solutions were in contact with the proboscis labellum from frame 30 to frame 40. Cell types imaged using a 2-photon microscope are Bract, Rattle, and Roundup.

## Calcium imaging with optogenetic activation of GRNs

For imaging responses in the Fdg cell type to optogenetic activation of GRNs, females of *13XLexAop2-IVS-p10-ChrimsonR-mCherry(attP18); Gr5a-LexA; 20XUAS-IVS-jGCaMP7b(VK00005)*, *13XLexAop2-IVS-p10-ChrimsonR-mCherry(attP18); 20XUAS-IVS-jGCaMP7b(attP5); ppk28-LexA*, or *13XLexAop2-IVS-p10-ChrimsonR-mCherry(attP18); 20XUAS-IVS-jGCaMP7b(attP5); Gr66a-LexA* were crossed to males of SS46913 (*Sterne et al., 2021*). For sugar and bitter integration experiments, virgins of a stock composed of either SS47082 (G2N-1) or SS47744 (Roundup) and *20xUAS-IVS-Syn21-Syt::Op-jGCaMP7b(attP18)* were crossed to males of *13XLexAop2-IVS-p10-ChrimsonR-mCherry(attP18); Gr5a-LexA::VP16(12-1)*;, *13XLexAop2-IVS-p10-ChrimsonR-mCherry(attP18);Gr5a-LexA::VP16(12-1)*;

*Gr66a-LexA*, or *13XLexAop2-IVS-p10-ChrimsonR-mCherry(attP18);;Gr66a-LexA*; and female progeny without balancers were selected for imaging. 2-photon imaging was performed as described above for imaging with taste stimulation, but 660 nm light was used to activate GRNs in place of direct stimulation of the proboscis with taste solutions. Two-second light pulses were delivered three times at 10 s intervals during imaging, and light was delivered through the objective in a widefield fashion under the control of a custom ScanImage plugin.

## Calcium imaging analysis

Image analysis was carried out in Fiji (*Schindelin et al., 2012*), CircuitCatcher (a customized Python program by Daniel Bushey *Dag et al., 2019*), Python, and R. First, in Fiji, Z stacks for each time point were maximum intensity projected and then movement corrected using the StackReg plugin with 'Rigid Body' or 'Translation' transformation (*Thévenaz et al., 1998*). Next, using CircuitCatcher, an ROI containing the neurites of the cell type of interest was selected along with a background ROI, and average fluorescence intensity for each ROI at each timepoint was retrieved. Then, in Python, background subtraction was carried out for each timepoint ($F_t$). To calculate $F_{initial}$, initial fluorescence intensity was calculated as the mean corrected average fluorescence intensity from frame 9–18 (for 1-photon imaging) or frame 0–19 (for 2-photon imaging and optogenetic imaging). Finally, the following formula was used to calculate ΔF/F: $F_t$-$F_{initial}$/$F_{initial}$. Area under the curve was approximated with the trapezoidal rule in Python using the NumPy.trapz function. Area under the curve was assessed from frames 20–25 (for 1-photon imaging), from frames 30–40 (for 2-photon imaging with taste stimulation), and from frames 15–18 (for 2-photon imaging with optogenetic activation).

## Quantification and statistical analysis

Statistical tests for behavioral assays were performed in Prism. For analysis of Proboscis Extension Response assays, Fisher's Exact Test was used in comparing the fraction of PER responses in experimental versus control flies. Statistical analysis of calcium imaging was carried out in R and Python. For imaging experiments carried out in a block design with three treatments, Quade tests were carried out in R using the PMCMRplus package (*Pohlert, 2021*). Quade test was chosen because it is more powerful than Friedman for a block-design experiment with three treatments (*Conover, 1999*). Other statistical analyses of calcium imaging were carried out in Python using the SciPy (*Virtanen et al., 2020*) and scikit-posthocs packages (*Terpilowski, 2018*).

## Acknowledgements

We thank Lori Horhor, Jolie Huang, Neil Ming, Vivian Nguyen, Andrea Sandoval, Neha Simha, Rivka Steinberg, and Parisa Vaziri for EM tracing contributions. We thank David T Harris for his initial studies on FMIn. We thank the FlyLight Project Team (https://www.janelia.org/project-team/flylight) for performing brain dissections, immunohistochemistry, and confocal imaging for split-GAL4 screening. Vivek Jayaraman provided unpublished fly lines used in this study. This work was supported by NIH R01DC013280 (KS), NIH F32DK117671 (GRS) and NIH F32DC018225 (PS). Neuronal reconstruction for this project took place in a collaborative CATMAID environment in which 27 labs are participating to build connectomes for specific circuits. Development and administration of the FAFB tracing environment and analysis tools were funded in part by National Institutes of Health BRAIN Initiative grant 1RF1MH120679-01 to Davi Bock and Greg Jefferis, with software development effort and administrative support provided by Tom Kazimiers (Kazmos GmbH) and Eric Perlman (Yikes LLC). Peter Li, Viren Jain and colleagues at Google Research shared automatic segmentation (*Li et al., 2019*). We acknowledge the Princeton FlyWire team and members of the Murthy and Seung labs for development and maintenance of FlyWire (supported by BRAIN Initiative grant MH117815 to Murthy and Seung). We thank Drs. Stefanie Hampel, Jinseop Kim, Mala Murthy, Andrew Seeds, Sebastian Seung, Ibrahim Tastekin and Rachel Wilson and members of their laboratories for FlyWire tracing. We thank K Eichler and members of the Connectomics Group in the Dept Zoology, University of Cambridge (G. Jefferis, M Costa) for FlyWire tracing and sharing some neurons ahead of publication. Confocal imaging for triple intersection characterization was conducted at the CRL Molecular Imaging Center, supported by the Helen Wills Neuroscience Institute and NSF DBI-1041078. We would also like to thank Holly Aaron and Feather Ives for their microscopy training and assistance. Members of the Scott lab provided comments on the manuscript.

# Additional information

## Funding

| Funder | Grant reference number | Author |
|---|---|---|
| National Institutes of Health | R01DC013280 | Kristin Scott |
| National Institutes of Health | F32DK117671 | Gabriella R Sterne |
| National Institutes of Health | F32DC018225 | Philip K Shiu |

The funders had no role in study design, data collection and interpretation, or the decision to submit the work for publication.

## Author contributions

Philip K Shiu, Gabriella R Sterne, Conceptualization, Resources, Formal analysis, Funding acquisition, Validation, Investigation, Visualization, Methodology, Writing - original draft, Writing – review and editing; Stefanie Engert, Conceptualization, Resources, Validation, Investigation, Visualization, Writing – review and editing; Barry J Dickson, Resources, Supervision, Writing – review and editing; Kristin Scott, Conceptualization, Resources, Supervision, Funding acquisition, Writing - original draft, Project administration, Writing – review and editing

## Author ORCIDs

Philip K Shiu ![ORCID] http://orcid.org/0000-0002-8794-5474
Gabriella R Sterne ![ORCID] http://orcid.org/0000-0002-7221-648X
Stefanie Engert ![ORCID] http://orcid.org/0000-0003-0644-8116
Kristin Scott ![ORCID] http://orcid.org/0000-0003-3150-7210

## Decision letter and Author response

Decision letter https://doi.org/10.7554/eLife.79887.sa1
Author response https://doi.org/10.7554/eLife.79887.sa2

# Additional files

## Supplementary files

• MDAR checklist

## Data availability

All data is included in the manuscript or available at https://catmaid-fafb.virtualflybrain.org.

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
