## [Editor Report]

The findings presented in this article contribute to a circuit-based understanding of how sweet and bitter tastes are integrated with the hunger state to drive feeding initiation in *Drosophila*. Anatomical, behavioral, and neuronal activity data support a multi-step pathway from sensory input to motor output. This manuscript, thus, advances our understanding of how multiple sensory cues are integrated with an internal state to reach a behavioral decision.

---

## [Decision Letter]

**Decision letter after peer review:**

Thank you for submitting your article "Taste quality and hunger interactions in a feeding sensorimotor circuit" for consideration by *eLife*. Your article has been reviewed by 3 peer reviewers, and the evaluation has been overseen by a Reviewing Editor and K VijayRaghavan as the Senior Editor. The following individual involved in review of your submission has agreed to reveal their identity: Simon G Sprecher (Reviewer #1).

Essential revisions:

All three reviewers were impressed by the large amount of high-quality data. All three support publication of this manuscript. Before a final decision can be made, a few points need to be addressed by editing the manuscript or improving presentation. No additional data are needed.

Please address all reviewer points in a point-by-point response and pay particular attention to improving clarity, background presentation, and result interpretation with the goal of clarifying the take-home messages for the readers.

*Reviewer #1 (Recommendations for the authors):*

The study is elegant and by themselves the two distinct parts – connectome and genetic neuron assessment- provide insight into the taste circuit organization. I do not have any major comments – or concerns- on the overall logic of reasoning or layout of the data. However, I do have a few points that I feel would be important to address.

While each of the two sections (connectome and neurogenetics) by itself is logical and experimentally quite well developed the link between the two sections is less clear. Much of the reasoning that underlies this connection is simply skipped or omitted. This should not be seen as a criticism of the data itself, but rather provide an option to be transparent with certainties and uncertainties. Along these lines I felt it would be important to know:

– How were "likely" sugar-sensing GRNs in the connectome identified? (line 77), can the authors explain the rationale behind this assumption?

– How certain is the identity linked between split-Gal4 lines and connectome lines? Are there putative other cells that could be mistaken for some of the GRN targets?

– Are the 15 second-order neurons all targets of the GRNs or which portion do they make up? What do the authors mean by "revealing that they represent 12 of the 13 cell types with the most synapses from candidate sugar GRNs" line 82/83.

– Less impacting, but still probably relevant: while the nomenclature of the neurons may be funny the rationale behind it is not explained (or not referred to). It may be worth to depict this in the method section.

As result of the quite extensive data presented, the core messages appear almost a bit forgotten. For instance, early redundancy in "necessity and sufficiency" for PER is to some degree discussed, but not further explored. What do the authors think is the relevance of this finding? How does it compare of what we know from direct EM connectomic analysis in other modalities adult fly or larva?

It is of course tempting to ask an array of questions concerning the functionality of the circuit beyond the current analysis. I am sure that the authors have considered this, and I do not feel any further experimental investigation is necessary.

However, there are a couple points that maybe worth discussing (again no data required):

– Do the authors have any idea or hypothesis of how the use of excitatory or inhibitory neurotransmitter may affect the proposed "circuit" function? How does this relate to satiation state?

– Do the authors have any idea about how isolated or connected the current circuit with other taste modalities is (it may be sufficient to discuss ideas on this)?

*Reviewer #3 (Recommendations for the authors):*

A couple of the experimental results seem counterintuitive and would benefit from further discussion:

(1) Optogenetic activation of a large subset of the individual neurons in the circuit (2nd order through pre-motor) is sufficient to drive PER, suggesting potential functional redundancy. But only a subset of 2nd order neurons (and none of the 3rd order or premotor neurons) appear required for PER in response to sugar. One might anticipate that inactivating neurons closer to the motor output might have a stronger impact on PER than inactivating upstream neurons. Does the presence of multiple motor outputs that contribute to PER downstream of the 2nd order neurons explain why inhibition of some of these upstream neurons have a more dramatic impact on the behavior? It would be useful to clarify the extent to which the analysis (and connectome) focuses on just one of the many motor neurons involved in stimulating PER in response to sugar.

(2) Another potentially confusing result is the response of the Usnea 2nd order neuron to water rather than sugar, despite its apparent connections to sugar-responsive gustatory neurons and its requirement for PER in response to sugar. What is the likely explanation for this result?

In terms of further speculation that would be of interest to readers:

(3) Do the authors have any speculative ideas about how/why the responses to G2N-1 and Clavicle activation are hunger-regulated?

(4) Do the authors have any thoughts about how Phantom is contributing to the response?

(5) The authors speculate the Fdg is responsive to GRNs from non-labellar tissue. Have they tested whether contact of the leg with sugar is sufficient to activate the Fdgs?

---

## [Author Response]

Reviewer #1 (Recommendations for the authors):The study is elegant and by themselves the two distinct parts – connectome and genetic neuron assessment- provide insight into the taste circuit organization. I do not have any major comments – or concerns- on the overall logic of reasoning or layout of the data. However, I do have a few points that I feel would be important to address.While each of the two sections (connectome and neurogenetics) by itself is logical and experimentally quite well developed the link between the two sections is less clear. Much of the reasoning that underlies this connection is simply skipped or omitted. This should not be seen as a criticism of the data itself, but rather provide an option to be transparent with certainties and uncertainties. Along these lines I felt it would be important to know:– How were "likely" sugar-sensing GRNs in the connectome identified? (line 77), can the authors explain the rationale behind this assumption?

Engert et al., 2022 defined sugar GRNs in the EM connectome by clustering GRNs using morphology and connectivity data and comparing the resulting clusters with immunostained GRNs responding to different taste categories (ln 444-447).

– How certain is the identity linked between split-Gal4 lines and connectome lines? Are there putative other cells that could be mistaken for some of the GRN targets?

The split-Gal4 lines are the best matches for the EM neurons, described in Methods (Identification of split-GAL4 lines from EM reconstructions). We now add (ln 543-548):

“Flywire dense reconstructions of second- and third-order neurons (Figure 1 —figure supplement 1F; Figure 2 —figure supplement 1B) identified no anatomically indistinguishable neurons, except for Bract 1 and Bract 2. Therefore, the high similarity in projection patterns between the split-Gal4 lines and the EM neurons, as well as the functional and behavioral responses of the split-Gal4 lines, provide confidence that the neurons labeled by the split-Gal4 lines match the EM neurons.”

– Are the 15 second-order neurons all targets of the GRNs or which portion do they make up?

The 15 second-order neurons are the top targets of GRNs. All targets with more than 40 synapses are shown in Figure 1 —figure supplement F. The 15 second-order neurons receive 21% of sugar GRN synaptic outputs. (ln 92)

What do the authors mean by "revealing that they represent 12 of the 13 cell types with the most synapses from candidate sugar GRNs" line 82/83.

We apologize for the confusion. As shown in Figure 1 —figure supplement 1F, of the 15 neurons with the most connections from sugar GRNs, only one neuron was not found in our CATMAID reconstruction analysis. However, of these 15 neurons, Usnea and Zorro are represented twice, because sugar GRNs synapse onto these neurons from both the right and left hemisphere. Thus, 12 of 13 cell types are found. To avoid this confusion, we have expanded the text to state, (ln 86-94),

“To assess the completeness of our second-order collection, we compared these 15 second-order neurons with the recently released Flywire dataset, a dense, machine learning based reconstruction of FAFB neurons (Dorkenwald et al., 2020; Eckstein et al., 2020). This comparison revealed that we identified 14 of the 15 neurons with the most synapses from sugar GRNs. These second-order neurons represent 12 unique cell types (Figure 1 —figure supplement 1F). The 15 second-order neurons we manually reconstructed account for 21% of sugar GRN synaptic outputs. We note that the distribution of second-order neurons has a very long tail, likely due in part to small neural fragments that are challenging to reconstruct.”

– Less impacting, but still probably relevant: while the nomenclature of the neurons may be funny the rationale behind it is not explained (or not referred to). It may be worth to depict this in the method section.

We have added a section in the methods (ln 460-466):

“Neuron Nomenclature

The vast majority of the neurons referred to here were named in Sterne et al., 2021. Zorro was named because the proximal neurite forms a “Z”. Scapula was named due to its resemblance to an inverted scapula bone, and Sternum was named due to its appearance, connectivity and proximity to Clavicle. Roundtree and Rounddown were named because they, like Roundup (named in Sterne et al., 2021), are premotor neurons.”

As result of the quite extensive data presented, the core messages appear almost a bit forgotten. For instance, early redundancy in "necessity and sufficiency" for PER is to some degree discussed, but not further explored. What do the authors think is the relevance of this finding?

We further add (ln 308-320):

“Each neuron elicits proboscis extension upon optogenetic activation, demonstrating that each neuron participates in a pathway for the behavior. Inhibiting activity of single second-order neurons reduced the behavioral response whereas inhibiting activity of third-order or premotor neurons did not. As inhibiting activity of single neurons did not abolish proboscis extension, this demonstrates that additional paths not requiring the individual neural cell type contribute to this innate behavior. These results are consistent with the circuit connectivity, which reveals that there are multiple routes between sugar GRNs and MN9 for proboscis extension. The multiple paths from second-order neurons to premotor neurons may enable proboscis extension to be recruited in different contexts to ensure robust feeding. More generally, multiple circuit paths may enhance behavioral flexibility by facilitating sensory tuning and multi-sensory integration (Miroschnikow et al., 2018, Ohyama et al., 2015).”

How does it compare of what we know from direct EM connectomic analysis in other modalities adult fly or larva?

We now note (ln 327-334):

“Consistent with our finding that there are multiple pathways for proboscis extension, EM circuit tracing of sensorimotor paths for feeding in *Drosophila* larvae revealed both direct GRN-motor neuron synapses as well as indirect pathways (Miroschnikow et al., 2018). The functional significance of this circuit organization has not been explored. Although we did not identify direct GRN-motor neuron synaptic connections for proboscis extension in adult *Drosophila*, other feeding subprograms such as pharyngeal pumping may utilize direct sensory to motor connections and remain to be discovered.”

It is of course tempting to ask an array of questions concerning the functionality of the circuit beyond the current analysis. I am sure that the authors have considered this, and I do not feel any further experimental investigation is necessary.However, there are a couple points that maybe worth discussing (again no data required):– Do the authors have any idea or hypothesis of how the use of excitatory or inhibitory neurotransmitter may affect the proposed "circuit" function? How does this relate to satiation state?

We now provide candidate neurotransmitter predictions for these neurons (Figure 2 —figure supplement 1D) (ln 161-163), generated by a machine learning classifier (Eckstein et al., 2020).

Further studies are required to examine neurotransmitters involved in satiation state.

– Do the authors have any idea about how isolated or connected the current circuit with other taste modalities is (it may be sufficient to discuss ideas on this)?

Yes, we now provide Figure 3 —figure supplement 1D, showing the GRN connections to the second-order neurons in the feeding initiation circuit. We also include additional discussion on this topic (ln 205-227):

“Given that two second-order neurons, Usnea and Phantom, responded to water taste stimulation despite synaptic connectivity to candidate sugar GRNs, we examined the connections of the second-order feeding initiation neurons from all GRNs of the right hemisphere (Engert et al., 2022). These GRNs have previously been clustered into candidate taste categories (sugar, water, bitter, high salt and low salt) based on their morphology and GRN-GRN connectivity. Remarkably, we found that the second-order neurons that receive sugar GRN inputs also receive inputs from candidate water and high-salt (ppk23-positive, Glut-positive) GRNs but do not receive inputs from candidate bitter or low salt (Ir94e) GRNs (figure 3 —figure supplement 1D). The connectivity is consistent with our calcium imaging studies showing responses to sugar taste detection but not bitter taste. However, responses to sugar and water were not consistent with the predicted connectivity for each neuron, suggesting the possibilities of state-dependence, network interactions, and/or errors in GRN modality categorization. As GRN category assignments in the EM dataset were based on anatomy and connectivity alone, some GRNs may be misclassified, leading to errors in assessing sensory inputs (Engert et el, 2022). While the connectivity suggests the exciting possibility that taste integration may occur at second-order neurons, further functional studies will be necessary to illuminate the taste categories that activate or inhibit individual second-order neurons under different nutritive states. Nevertheless, our studies demonstrate that most second-order neurons respond to sugar taste stimulation but not water or bitter tastes.”

Reviewer #3 (Recommendations for the authors):A couple of the experimental results seem counterintuitive and would benefit from further discussion:(1) Optogenetic activation of a large subset of the individual neurons in the circuit (2nd order through pre-motor) is sufficient to drive PER, suggesting potential functional redundancy. But only a subset of 2nd order neurons (and none of the 3rd order or premotor neurons) appear required for PER in response to sugar. One might anticipate that inactivating neurons closer to the motor output might have a stronger impact on PER than inactivating upstream neurons. Does the presence of multiple motor outputs that contribute to PER downstream of the 2nd order neurons explain why inhibition of some of these upstream neurons have a more dramatic impact on the behavior?

Yes, this is a possibility. We speculate the premotor neurons, Roundup, Rounddown and Roundtree, as well as others that we may not have identified, may redundantly contribute to proboscis extension. More generally, we have added these comments on the behavioral activation and silencing results (ln 308-320):

“Each neuron elicits proboscis extension upon optogenetic activation, demonstrating that each neuron participates in a pathway for the behavior. Inhibiting activity of single second-order neurons reduced the behavioral response whereas inhibiting activity of third-order or premotor neurons did not. As inhibiting activity of single neurons did not abolish proboscis extension, this demonstrates that additional paths not requiring the individual neural cell type contribute to this innate behavior. These results are consistent with the circuit connectivity, which reveals that there are multiple routes between sugar GRNs and MN9 for proboscis extension. The multiple paths from second-order neurons to premotor neurons may enable proboscis extension to be recruited in different contexts to ensure robust feeding. More generally, multiple circuit paths may enhance behavioral flexibility by facilitating sensory tuning and multi-sensory integration (Miroschnikow et al., 2018, Ohyama et al., 2015).”

It would be useful to clarify the extent to which the analysis (and connectome) focuses on just one of the many motor neurons involved in stimulating PER in response to sugar.

We have added the following section (ln 136-139):

“Proboscis motor neurons 4, 6, 7 and 9 are involved in extending different segments of the proboscis for feeding initiation (McKellar et al., 2020). We focused on the well-studied motor neuron 9 (MN9), which is necessary and sufficient for extension of the rostrum, the largest portion of the proboscis (Gordon and Scott; McKellar et al., 2020).”

(2) Another potentially confusing result is the response of the Usnea 2nd order neuron to water rather than sugar, despite its apparent connections to sugar-responsive gustatory neurons and its requirement for PER in response to sugar. What is the likely explanation for this result?

We agree that it is confusing that Usnea receives sugar GRN inputs but does not respond to sucrose. We hypothesize that there is a state-dependency to the response that we have not identified. We now discuss GRN-second order connectivity in detail (ln 205-227).

In terms of further speculation that would be of interest to readers:(3) Do the authors have any speculative ideas about how/why the responses to G2N-1 and Clavicle activation are hunger-regulated?

We are unable to speculate on this, as many modulators have been implicated in feeding regulation. Further study is required to determine the hunger signals that modulate G2N-1 and Clavicle.

(4) Do the authors have any thoughts about how Phantom is contributing to the response?

Phantom is predicted to be inhibitory (New Figure 2 —figure supplement 1D). Phantom synapses onto Scapula (Figure 5A), and Scapula likely inhibits premotor neurons. Thus, we speculate that Phantom activation may promote premotor activity by relieving inhibition on premotor neurons.

(5) The authors speculate the Fdg is responsive to GRNs from non-labellar tissue. Have they tested whether contact of the leg with sugar is sufficient to activate the Fdgs?

We have not seen responses of Fdg to stimulation of the leg with sugar (data not shown). However, optogenetic activation of the entire subset of GRNs covered by Gr5a-LexA elicits calcium responses in Fdg. Therefore, activation of pharyngeal GRNs alone or simultaneous activation of some combination of labellar, pharyngeal, and leg GRNs may be required to activate Fdg.